# A New Slurry for Photocatalysis-Assisted Chemical Mechanical Polishing of Monocrystal Diamond

Junyong Shao [1,2], Yanjun Zhao [1,2], Jianhui Zhu [1], Zewei Yuan [3,*], Haiyang Du [3] and Quan Wen [4,*]

1   State Key Laboratory for High Performance Tools, Zhengzhou Research Institute for
    Abrasives & Grinding Co., Ltd., Zhengzhou 450001, China; shaoyong102@139.com (J.S.); zyj@zzsm.com (Y.Z.)
2   School of Materials Science and Engineering, Zhengzhou University, Zhengzhou 450001, China
3   School of Mechanical Engineering, Shenyang University of Technology, Shenyang 110870, China
4   School of Mechanical Engineering & Automation, Northeastern University, Shenyang 110819, China
*   Correspondence: zwyuan@sut.edu.cn (Z.Y.); wenquan@me.neu.edu.cn (Q.W.)

**Abstract:** Diamond needs to have a perfectly smooth surface due to the growing requirements in the fields of electronic semiconductors, optical windows and high-fidelity loudspeakers. However, the polishing of diamonds is highly challenging due to their exceptional hardness and chemical stability. In this study, a new polishing slurry is prepared for the proposed photocatalysis-assisted chemical mechanical polishing (PCMP) approach to obtain an ultra-smooth surface for large-area diamond. The analyses and experimental findings revealed the significance of the photocatalyst, abrasive, electron capture agent and pH regulator as essential components of the PCMP slurry. $TiO_2$ with a 5 nm pore size and P25 $TiO_2$ possess improved photocatalysis efficiency. Moreover, diamond removal is smooth under the acidic environment of $H_3PO_4$ due to the high oxidation–reduction potential (ORP) of the slurry, and, during the methyl orange test, P25 $TiO_2$ exhibits reasonable photocatalytic effects. Moreover, in 8 h, a smooth surface free of mechanical scratches can be obtained by reducing the surface roughness from Ra 33.6 nm to Ra 2.6 nm.

**Keywords:** photocatalysis; chemical mechanical polishing; diamond; slurry; preparation


## 1. Introduction

Large-area diamonds are gaining popularity as a new generation of semiconductor materials due to their outstanding physical, thermal, optical and chemical properties, such as high hardness, strong chemical inertness, excellent thermal conductivity, high elasticity modulus, large electrical resistance, broad electronic gap, wide-range transparency and a small friction coefficient [1]. Particularly, the development of chemical vapor deposition (CVD) technology has overcome the restriction of rare, expensive and small-sized natural diamond, and it has greatly expanded the fields of application for diamonds from the traditional jewels and tools to electronic semiconductors, optical windows, high-fidelity loudspeakers, high-energy accelerators, etc. [2–4]. Diamond's unique qualities have the potential to significantly boost these devices' functionality. To meet the criteria of the aforementioned applications, however, the rough surface of diamonds developed by the CVD method is challenging [5]. As a result, an ultra-high-precision process for large-area diamonds is required for effective application.

Diamond has a high degree of hardness and chemical stability, making it difficult to smooth its surface using conventional processing methods. The polishing of diamonds has received a great deal of attention from researchers, who have also presented a variety of techniques to improve surface quality and material removal rates [6]. These polishing techniques can be sorted into five categories based on how the material is removed: micro-chipping, diamond to graphite conversion, evaporation, sputtering and chemical reaction [7].

Micro-chipping happens at the interface of two contact surfaces when the friction force is larger than the binding force of the materials. Mechanical polishing and grinding both use micro-chipping mechanisms to remove diamond [8,9]. Cracks typically develop on a diamond's surface during the polishing process. Additionally, when the abrasive size is less than 1 um, the rate of material removal is relatively low, which can reduce the strength and surface roughness of a polished workpiece.

The second removal method involves the conversion of diamond to graphite. Diamond is readily converted to graphite because of its thermodynamically metastable structure, and the metals of iron, stainless steel, cerium, nickel, manganese and titanium are efficient catalysts for the process [10,11]. To prepare a polishing plate or grinding wheel, thermochemical polishing and tribochemical polishing typically use these metals and alloys to achieve a high removal rate [12,13].

A non-contact technique for polishing curved surfaces and localized tiny areas is laser polishing [14]. Due to the laser's energy, diamond is extracted through a form of evaporation. After laser polishing, the diamond's surface will still have a layer of graphite or non-diamond carbon that can be removed with chemical mechanical polishing or mechanical polishing [15].

In addition to laser polishing, sputtering and plasma can also be used for the non-contact polishing of diamond [16]. Ion beam polishing uses an ion source for polishing sputtering. The carbon atoms at the stroke surface are bombarded with ions [17,18]. Complex forms can be polished by ion beam polishing; however, sample size is constrained by the ion beam and chamber size.

Many researchers believe that chemical reactions during the polishing process are advantageous since they can shift the rate at which the material is removed, even though diamond has good chemical inertness. Common methods for polishing diamond with chemical reaction mechanisms include reactive ion etching and chemical mechanical polishing. $O_2$ and $H_2$ gases react with diamonds during reactive ion etching [19]. Although it is faster than ion beam sputtering, the diamond surface may become contaminated because of plasma heating. The cost is relatively significant, and the plasma size restricts the sample size. Chemical mechanical polishing (CMP) is an effective method to achieve atomic surface globally, and, therefore, the most promising technique for large-area diamonds is chemical mechanical polishing [20,21]. To extract the diamond atom by chemical reaction or tribochemical reaction, it employs either molten mixed salt containing $NaNO_3$, $KNO_3$ and KOH or produced slurry with strong oxidants containing $CrO_3$, $KMnO_4$, $H_2O_2$ and $K_2Cr_2O_7$ [22]. However, the traditional CMP slurry usually contains toxic and corrosive ingredients, resulting in pollution to the environment [23].

To overcome this challenge of traditional slurry and increase the material removal rate, strong oxidants, such as $K_2FeO_4$ and Fenton reagent, are used to prepare polishing slurry [24]. These studies are a great contribution to the conventional CMP and manufacturing, effectively eliminating the pollution to the environment [25,26]. However, the polishing effect might be negatively impacted by storage failure of this type of polishing slurry with a strong oxidant. Waste liquid treatment and recovery are highly expensive, and discharging waste liquid after polishing can seriously harm the environment and humans. Furthermore, the polishing parameters can be used to actively alter the mechanical action in the polishing process, but the chemical action is much more difficult to control. Given this, the current study proposes a photocatalysis-assisted chemical mechanical polishing (PCMP) approach to accomplish the ultra-smooth polishing of large-area diamonds via the production of a polishing slurry for the proposed PCMP.

## 2. Principle, Methods and Experiments

### 2.1. Preparation of PCMP Slurry

Photocatalysis is a new oxidant technique developed recently to use solar energy for environmental purification and energy purification [27]. As shown in Figure 1, on the surface of the $TiO_2$ nanoparticles, which are exposed to ultraviolet irradiation, many

holes and hydroxyl radical ·OH are formed [28]. The produced holes and hydroxyl radical ·OH with a strong oxidizability on surface of $TiO_2$ nanoparticles are easy to react with diamond to achieve the goal of diamond removal at the atomic level. The mechanical action of abrasive during the polishing process increases the atomic activity of diamond surface material (changing to be activated diamond carbon C*), which shifts the pace of the chemical reaction between diamond and oxidant in slurry. As a result, the rate of material removal gradually rises. According to the photocatalysis oxidation mechanism, the PCMP method is proposed as in Figure 2. According to the requirement of polishing process, the oxidizability of slurry can be adjusted in terms of the ultraviolet (UV) light powder and $TiO_2$ concentration. The slurry is safe for humans and the environment because it contains no toxic components. The hardness of titanium dioxide and the smaller size of abrasive particles as compared to diamonds enable the ultra-smooth polishing of large-area diamonds.

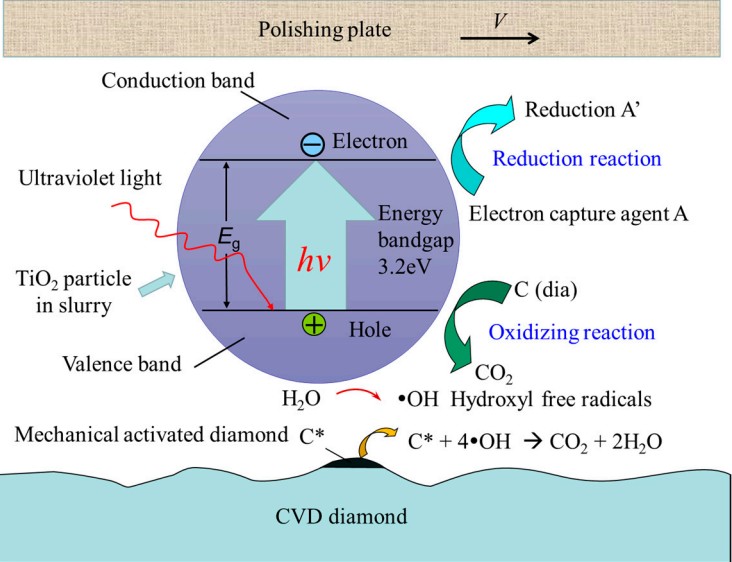

**Figure 1.** Schematic diagram of photocatalysis oxidation mechanism.

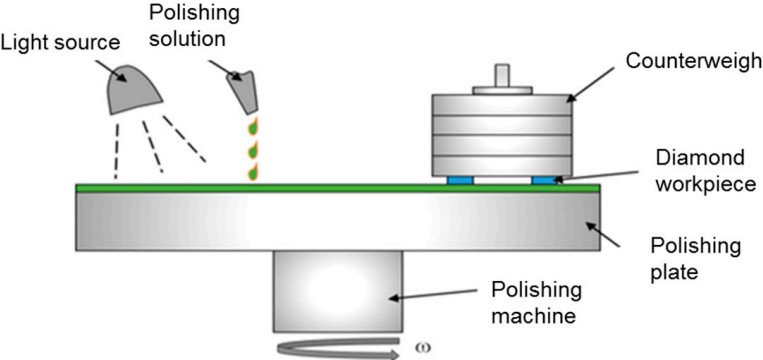

**Figure 2.** Schematic diagram of PCMP process.

The appropriate slurry preparation for the PCMP is of utmost importance. The effectiveness of a photocatalytic reaction mostly depends on the separation of electrons and holes, the rate of migration to the catalyst surface, the rate of oxidation and reduction caught by reactants, catalyst particle size, crystal type, UV light source, pH value, oxidant, etc. According to the photocatalysis principle, the photocatalyst, abrasive, electron capture agent, pH regulator and dispersant need to be incorporated in the photocatalysis-assisted chemical mechanical polishing slurry as shown in Table 1.

**Table 1.** Components of PCMP slurry.

| Photocatalyst | Abrasive Materials | Electron Trapping Agent | pH Regulator |
|---|---|---|---|
| 5 nm $TiO_2$ | $Al_2O_3$ | $H_2O_2$ | NaOH |
| 10 nm $TiO_2$ | diamond | $K_2FeO_4$ | $H_3PO_4$ |
| 20 nm $TiO_2$ | SiC | | |
| P25 $TiO_2$ | | | |
| ZnO | | | |

Alumina, silicon carbide and diamond abrasives are a few examples of super-hard abrasives that are typically used to polish diamond because of their high degree of hardness. This compares the polishing effectiveness of the abrasives of alumina and diamond. Diamond abrasives are the most hard abrasives, which are usually used in the process of lapping or mechanical polishing. They are not suitable for CMP and PCMP because diamond grits in slurry can be oxidized along with diamond workpiece. Additionally, diamond abrasives are easy to cause scratches on workpiece. The material removal rate will significantly decline with the decrease in diamond abrasive size. Therefore, the diamond abrasives are used as the abrasive of mechanical lapping for diamond rough treatment. Alumina abrasives are used in PCMP slurry to verify the effect of chemical action in PCMP on the material removal of diamond in this study because they cannot be oxidized by the oxidant in slurry.

The most frequently employed photocatalysts are nanoparticles of $TiO_2$. Anatase, rutile and brookite are the three crystal types of titanium dioxide. Rutile titanium dioxide has a lower redox potential (3.03 eV) than anatase (3.2 eV) and, as a result, possesses a weaker redox capability [29]. When the nano $TiO_2$ particles are exposed to UV light, the electrons in the valence band may be stimulated to move into the conduction band, which results in generation of holes in the valence band. As a result, semiconductor particles produce electron ($e^-$) and hole ($h^+$) pairs [30]. With the band gap $E_g$ as fallowing, the required wavelength $\lambda_g$ of UV light can be calculated [31].

$$\lambda_g(nm) = \frac{1240}{E_g(eV)}, \tag{1}$$

To enable the generation of electrons and holes on the surface of titanium dioxide particles, the UV light wavelength should, according to the Equation (1), be less than 387.5 nm [31]. The photo-generated holes have a standard redox potential of 3.2 V, which is significantly higher than the redox potentials of popular oxidants such as ozone (2.07 V), potassium ferrate (2.2 V), potassium permanganate (1.7 V) and chlorine (1.36 V); they have a high oxidizability. The hydroxyl radical ·OH (the standard redox potential is 2.76 V) can be produced by oxidizing the OH- and $H_2O$ adsorbed on the surface of $TiO_2$ particles [32,33]. Strong oxidizability exists for the hydroxyl radical. By oxidizing carbon in the surface of diamond, it can remove diamond. Through the use of material combination simulation technologies and X-ray photoelectric emission experiments, many researchers found that the mixed crystal titanium dioxide is superior to single-crystal titanium dioxide [34]. As photocatalyst, Degussa Company's nano P25 mixed crystal $TiO_2$ and single-crystal anatase $TiO_2$ are used in the experiments. P25 $TiO_2$ particles are typically about 25 nm in size. The ratio of anatase to rutile in BET is roughly 80:20, and the surface area is 50 $m^2/g$.

It is crucial to add an appropriate amount of an electron capture agent to the PCMP process to prevent the recombination of holes and electrons. This enhances photocatalytic reaction by allowing numerous holes to directly participate in oxidation reaction or to indirectly generate hydroxyl radical ·OH. Strong oxidizers that are non-toxic, odorless and inexpensive are $H_2O_2$ and $K_2FeO_4$. During the reaction process, they do not produce any substances that are harmful to humans. Therefore, the preferred electron capture agents in this investigation are $H_2O_2$ and $K_2FeO_4$.

To increase the oxidizability and the dispersibility of the PCMP slurry, additional ingredients, such as an abrasive, an electron capture agent, a pH regulator and dispersant, are added. Phosphoric acid can be adopted for chemical mechanical polishing of diamond due to its non-volatilization and decomposition characteristics. In light of this, phosphoric acid was utilized in the study to regulate pH.

### 2.2. Characterization of PCMP Slurry

As illustrated in Figure 3, an oxidation characterization test was used to determine the oxidation–reduction potential (ORP), pH value and conductivity of the PCMP slurry. The ORP was measured with AZ86505 oxidation–reduction potentiometer (AZ Instrument Corporation, Taiwan, China). The ORP values were recorded when different photocatalyst, electron capture agent and pH regulator were added into solutions. In addition to these parameters, the methyl orange degradation test can be used to determine the oxidizability of PCMP slurry. Methyl orange ($C_{14}H_{14}N_3SO_3Na$) is a water-soluble dye that comes in orange powder form [35]. Figure 4 shows the experiment setup for oxidation of methyl orange. Methyl orange will be oxidized by the hydroxyl radical ·OH formed on the surface of $TiO_2$ particles when subjected to UV radiation. As a result, the orange slurry will fade out. The PCMP slurry's ability to oxidize can be indicated by fade time. Higher oxidation resistance results from a shorter fading time.

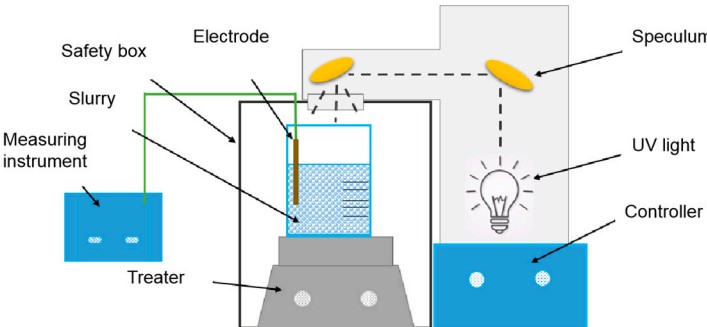

**Figure 3.** Schematic diagram of oxidation characterization test.

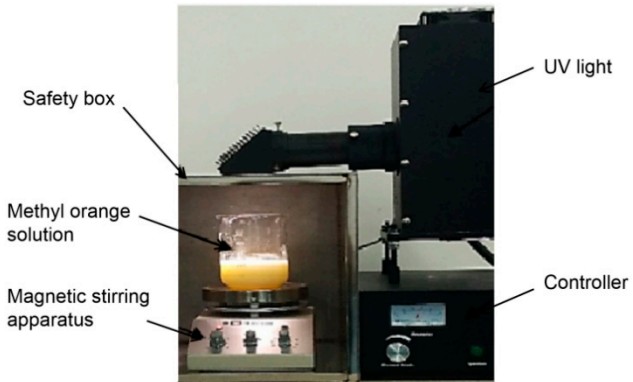

**Figure 4.** Experiment apparatus for oxidation of methyl orange.

### 2.3. PCMP Experiments

The UNIPOL-1202 automatic lapping and polishing equipment was used for PCMP studies as shown in Figure 5. Aluminum oxide is used as the polishing plate. Three CVD diamond workpieces are pasted with paraffin to the base of the polishing head during the polishing operation. Changing the number of the counterweight allows for adjustment of the polishing pressure. Before PCMP, rough asperities on the diamond workpieces must be removed by stepwise lapping them with 3–6 μm, 1–3 μm, 0.5–1 μm and 0–0.5 μm diamond abrasives. The slurry for PCMP contains 0.5 g of P25 $TiO_2$ particles, 6 g of aluminum oxide

abrasives and 3 mL of $H_2O_2$ in every 100 mL water. The polishing rotational speed and pressure are 60 r/min and 1.09 MPa, respectively.

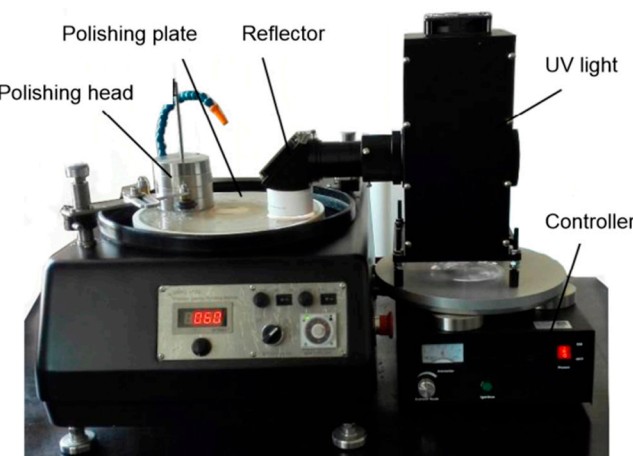

**Figure 5.** Experiment apparatus for oxidation of methyl orange.

Since nano $TiO_2$ particles can only absorb UV light with wavelength smaller than 387.5 nm, ultraviolet light is crucial for PCMP. At the moment, the photocatalytic oxidation experiment mostly uses UV light, such as xenon lamp, deuterium lamp, bromine tungsten lamp, etc. The Merc-1000 W mercury lamp served as the light source for this work. A mercury lamp can provide light with a maximum power of 1500 W, as shown in Figure 6. The Merc-1000 W mercury lamp's current may be adjusted between 5 A and 25 A and air cooling is the primary cooling method for this type of lamp. During PCMP process, different filters can be installed to control the light wavelength.

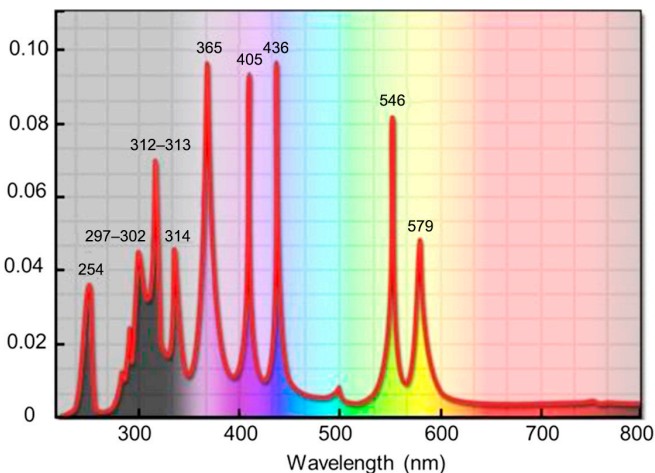

**Figure 6.** Merc-1000 W mercury lamp light source with the spectra.

### 3. Results and Discussion

*3.1. Oxidation–Reduction Potential of PCMP Slurry*

The oxidizability of a solution can be assessed using the ORP value, and, in water, it varies with different photocatalysts, as illustrated in Figure 7. The ORP values of 5 nm $TiO_2$ are the highest among various photocatalysts and display an obvious increase when the slurry is placed under the UV light. The ORP values of 10 nm $TiO_2$, P25 $TiO_2$ and ZnO are smaller than 5 nm $TiO_2$ but larger than 20 nm $TiO_2$ distinctly, indicating that the photocatalytic performance is not only affected by the size of photocatalyst particles but also is related to the crystal form of photocatalysts. P25 is a type of mixed crystal $TiO_2$. It possesses higher ORP values than 20 nm $TiO_2$, although the typical size of P25 $TiO_2$ particles is about 25 nm. The lowest ORP is observed in pure water because of its low oxidizability.

Additionally, the ORP values display a remarkable increase when $H_2O_2$ is added to the solution. This is owing to the fact that the $H_2O_2$ increases the conductivity of the solution and prevents the recombination of electrons and holes due to its high oxidizability.

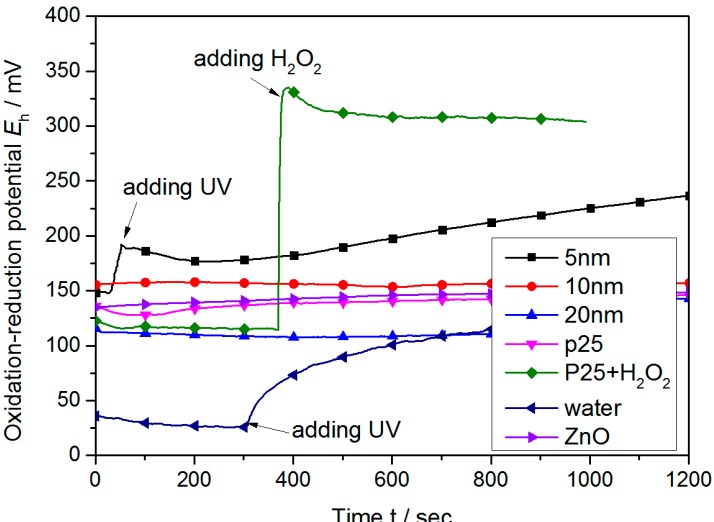

**Figure 7.** ORP varying with different photocatalysts.

Figure 8 illustrates how ORP varies with $TiO_2$ concentration in the presence of UV radiation. ORP increases as $TiO_2$ concentration increases because more hydroxyl radicals are produced on the surface of $TiO_2$ particles, which also increases the conductivity of the slurry. The conductive solution is beneficial for the measurement of ORP. However, too many $TiO_2$ particles in a slurry will decrease the absorption of UV light due to the slurry becoming non-transparent when there are many $TiO_2$ particles. That is why the increase in ORP is not obvious when the concentration of $TiO_2$ particles is high, especially when $H_2O_2$ and $H_3PO_4$ are already in the slurry. Additionally, it should be noted that the addition of $H_2O_2$ and $H_3PO_4$ can significantly increase the oxidizability of the slurry. The introduction of an acidic condition is useful to increase the oxidizability of a slurry. It should be noted that the gas film produced on the surface of an electrode will affect the measurement of ORP when $H_2O_2$ and $H_3PO_4$ are added into the solution. The measured ORP values will be smaller than the actual ORP values of a particle area.

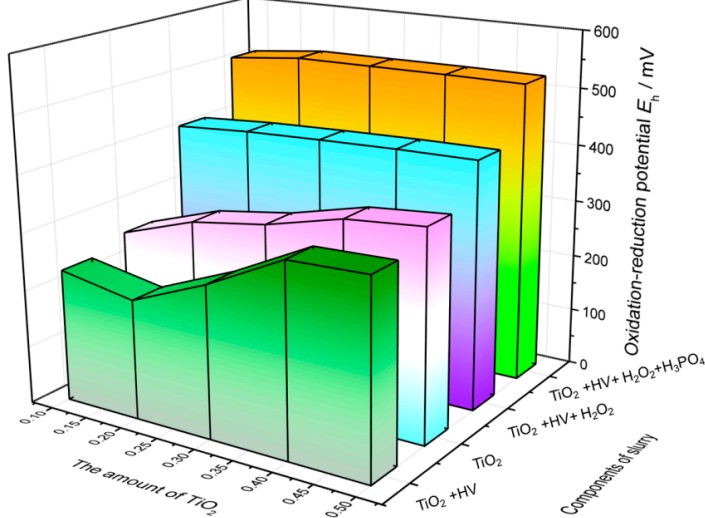

**Figure 8.** ORP varying with the amount of $TiO_2$.

The electrons and holes may readily recombine during the photocatalytic reaction process. The recombination probability of electron and hole can be decreased by adding

an appropriate quantity of oxidant that captures an adequate number of electrons. As a result, many holes can oxidize water and produce a hydroxyl radical. The holes and hydroxyl radical generated can oxidize the diamond and speed up the photocatalytic activity. Oxidants are typically chosen as electron capture agents. In the present study, $H_2O_2$ and $K_2FeO_4$ are utilized as electron capture agents due to their safety for the environment and human body. Figure 9 shows that the oxidizability of the slurry increases as the amount of the electron capture agent increases. Due to its better chemical stability, $H_2O_2$ outperforms $K_2FeO_4$, whereas potassium ferrate can be easily decomposed and lose its effectiveness.

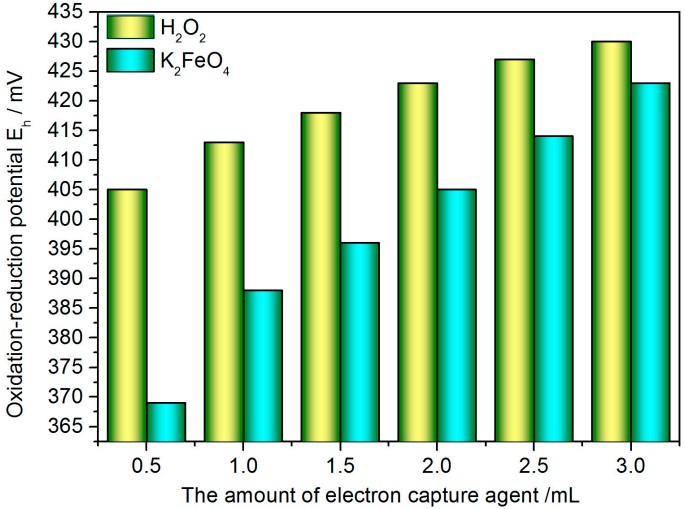

**Figure 9.** ORP varying with the amount of electron capture agent.

The ORP value is measured while the slurry is dipped into various concentrations of $H_2O_2$ to assess the stability of the mixture. According to Figure 10, if $H_2O_2$ is added to the slurry, the ORP rises significantly, indicating that the slurry's oxidizability has significantly increased. After some time, however, the ORP and oxidizability of the slurry start declining. Adding too much $H_2O_2$ will not only initiate the slurry's decomposition but also prevent ORP from increasing further. The ORP value essentially approaches 310 mV after 1.0 mL of $H_2O_2$ is added into the slurry. Due to its improved stability with the addition of 0.5 mL $H_2O_2$, the slurry's ORP value remains high after 500 s. Therefore, during the polishing process, an appropriate quantity of $H_2O_2$ must be intermittently added to the slurry to maintain the slurry's activity.

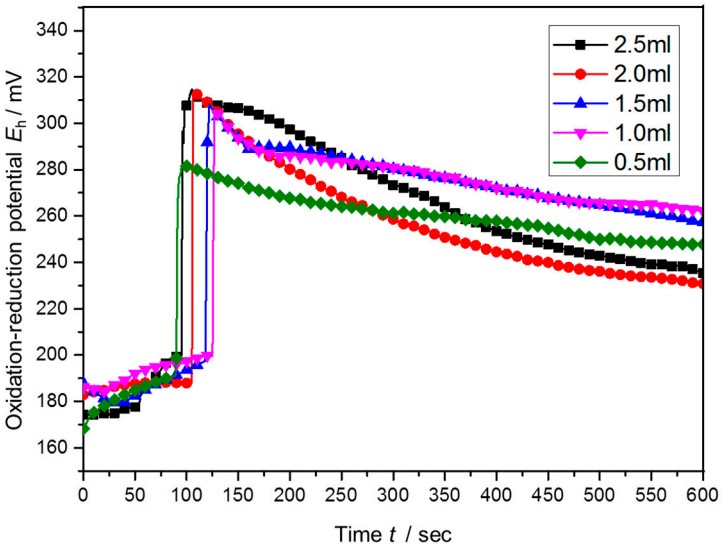

**Figure 10.** ORP of slurry varying with irradiation of UV light.

*3.2. pH Value of PCMP Slurry*

The stability and oxidation efficiency of the polishing solution have a major effect on pH in chemical mechanical polishing. The pH value will affect several characteristics, including the position of the valence band and conduction band, the surface adsorption of the functional group and the aggregation of $TiO_2$ catalyst. Large amounts of OH- are present in alkaline solutions with high pH levels, which encourages the transfer of holes from the interior of $TiO_2$ particles to the surface. On the contrary, the acidic solution's low pH makes the surface of $TiO_2$ easily protonated and subsequently positively charged, which is favorable for the transfer of photogenerated electrons to the surface of the $TiO_2$ particles. In the chemical mechanical polishing experiments, phosphoric acid was used as a pH regulator, making it hard to volatilize and decompose. The pH and ORP of the $TiO_2$–$H_2O_2$ slurry vary with the amount of phosphoric acid, as indicated in Figure 11. The pH value steadily drops from 2.26 to 1.66 with a rise in phosphoric acid. The ORP value was significantly increased initially by the addition of 0.2 mL phosphoric acid, reaching 501 mV. However, as more phosphoric acid was added, the growth rate gradually decreased, indicating that $H_2O_2$ in the photocatalytic polishing solution can capture electrons.

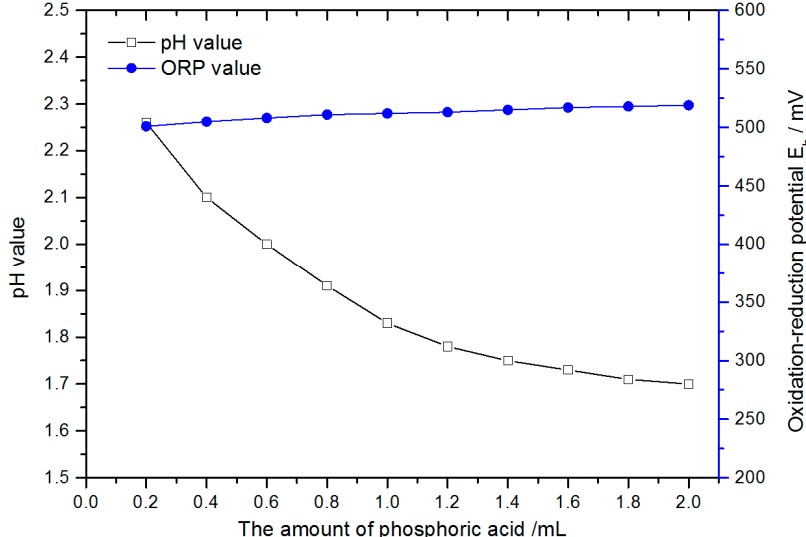

**Figure 11.** pH and ORP of slurry varying with the amount of phosphoric acid.

Through UV photocatalysis reaction, the diamond powder can oxidize its carbon into CO or $CO_2$ when combined with slurry [36,37]. In slurry, $CO_2$ dissolves and takes the forms $H_2CO_3$, $HCO_3^-$ and $CO_3^{2-}$. There are two equations for dissociation equilibrium:

$$2 \cdot OH + C = CO\uparrow + H_2O \tag{2}$$

$$4 \cdot OH + C = CO_2\uparrow + 2H_2O \tag{3}$$

$$H_2CO_3 \leftrightarrow HCO_3^- + H^+, \ K_1 = 4.2 \times 10^{-7}, \tag{4}$$

$$HCO_3^- \leftrightarrow CO_3^{2-} + H^+, \ K_2 = 5.6 \times 10^{-11}, \tag{5}$$

The primary reaction product in a solution with a pH of 6~7 is bicarbonate, which has an alkaline pH. As a result, the pH of the slurry steadily increases as the number of diamond abrasives increases. Additionally, the presence of diamond abrasive reduces the transmission of UV light in a solution. It also slows down the rate at which a photocatalyst decomposes. The chemical reaction impacts only a small portion of the diamond particles, but a consistent downward trend can be observed for ORP, as depicted in Figure 12.

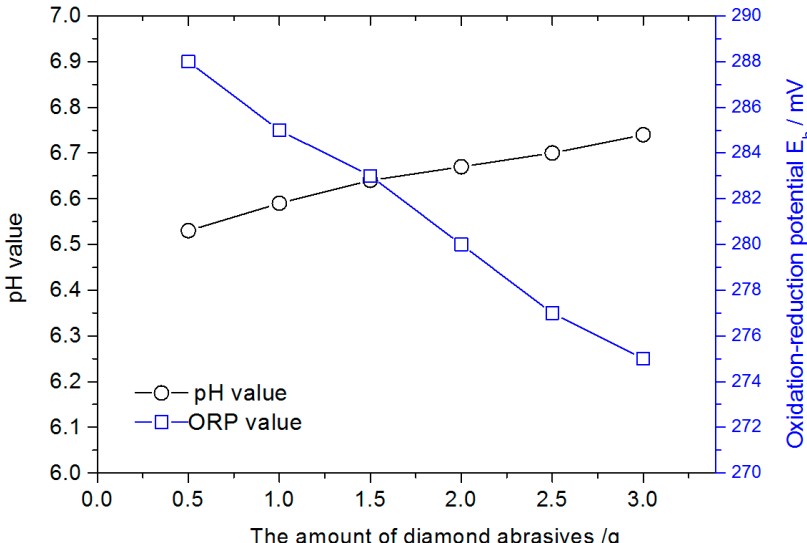

**Figure 12.** pH and ORP of slurry varying with the amount of diamond abrasives.

### 3.3. Conductivity of PCMP Slurry

Nano $TiO_2$ particles are conductive. Therefore, the slurry conductivity will change when nano $TiO_2$ particles are added to the slurry. Figure 13 illustrates how the conductivity of the solution varies as different quantities of $TiO_2$ are added into 50 mL water. The conductivity of a solution seems to be improved by the addition of a catalyst. When the slurry with nano $TiO_2$ particles is placed under the radiation of UV light, the conductivity of the slurry cannot change obviously no matter how many nano $TiO_2$ particles are added. It indicates that the photocatalyzed reaction usually occurs on the surface of nano $TiO_2$ particles, which do not affect the conductivity of a slurry. The conductivity of a slurry is mainly affected by the amount of nano $TiO_2$ particles. Additionally, the pH of a slurry continuously increases after adding nano $TiO_2$ particles. It implies that the slurry undergoes some chemical reaction. It should be noted that excess of $TiO_2$ powder will reduce the absorbance of UV light, and that will decrease the efficiency of the photocatalytic reaction.

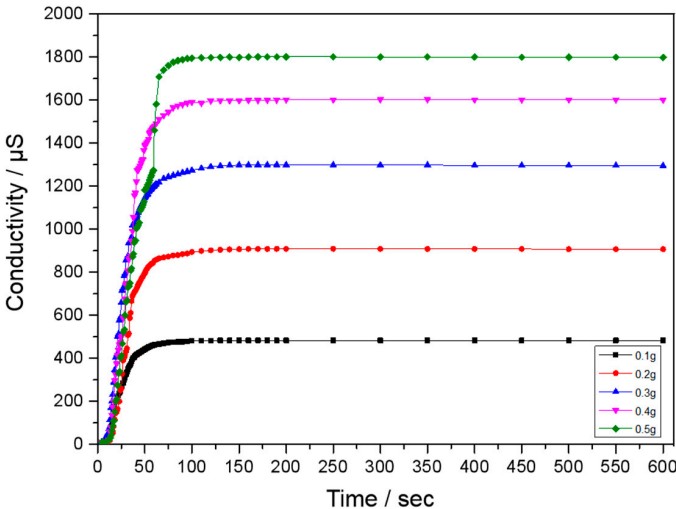

**Figure 13.** The conductivity varying with the amount of $TiO_2$.

Figure 14 shows the conductivity of the solution under UV irradiation for 10 min when 0.2 g of $TiO_2$ is added in various amounts of $H_2O_2$. By increasing the $H_2O_2$ dosage, the conductivity of the solution increases in the same reaction time. As $H_2O_2$ decomposes after being exposed to UV light for 100 s, the conductivity of the solution gradually falls. Generally, the additional $H_2O_2$ takes part in the capture of photogenerated electrons while

reacting with holes on the surface of the TiO$_2$ particles to produce peroxides. Consequently, the initial stage has more holes, which improves the photocatalytic activity and conductivity. The activity and conductivity of slurry, however, decreased by the excess reaction of H$_2$O$_2$ and holes. Therefore, the photocatalytic activity and the material removal benefit from the addition of a suitable amount of H$_2$O$_2$ to the slurry.

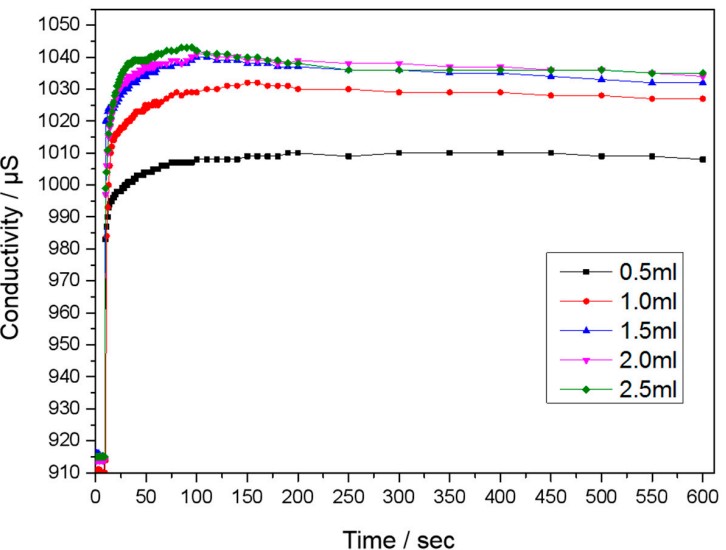

**Figure 14.** The conductivity varying with the amount of H$_2$O$_2$.

### 3.4. Oxidation Test of Methyl Orange

Methyl orange (C$_{14}$H$_{14}$N$_3$SO$_3$Na) was added to the polishing slurry to better characterize the slurry's oxidizability. For the test, methyl orange in orange–yellow powder form is used. The yellow color will quickly degrade if the methyl orange is oxidized by the slurry's oxidant. Therefore, the rate of yellow color degradation can be utilized to assess the oxidizability of the slurry.

The slurry for the photocatalytic oxidation test was made by adding 200 mL of methyl orange solution with 0.5 g of P25 TiO$_2$, 1.5 mL of H$_2$O$_2$ and a small amount of sodium hexametaphosphate. To guarantee the homogeneous dispersion of the solution, the slurry underwent a 20 min ultrasonic dispersion treatment. After that, the slurry was subjected to UV light. When methyl orange is added, the initial solution appears white and subsequently turns yellow, as illustrated in Figure 15. The methyl orange solution is discolored after 30 min of UV exposure. Figure 15d depicts that the methyl orange solution recovers to its original color and is entirely degraded after being exposed to ultraviolet light for 60 min.

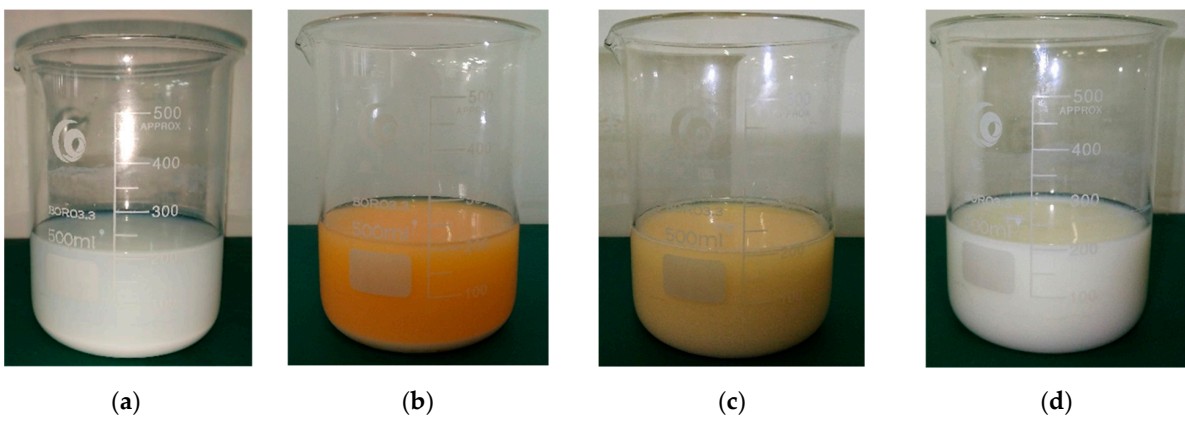

(a)            (b)            (c)            (d)

**Figure 15.** The color change of solution in photocatalytic oxidation test. (**a**) Initial solution; (**b**) adding methyl orange; (**c**) irradiation for 30 min; (**d**) irradiation for 60 min.

According to the analysis of the experimental results and the semiconductor photocatalytic oxidation mechanism, after exposure to UV light, a significant number of photogenerated holes are created in the solution and on the surface of the photocatalyst, and hydroxyl radical ·OH is subsequently produced on the surface of $TiO_2$ particles. Photogenerated holes and the hydroxyl radical ·OH both demonstrate considerable oxidation [20]. In the photocatalytic reaction, organic compounds such as methyl orange that are adsorbed on the surface of $TiO_2$ particles and in the solution will react with photogenerated holes and hydroxyl radical ·OH, causing a portion of the methyl orange to be directly degraded into $CO_2$ and $H_2O$. This leads to the decolorization phenomenon, as depicted in Figure 15.

### 3.5. Polishing Diamond with PCMP Slurry

Figure 16 shows that the diamond surface roughness varies according to the polishing slurries. Before the PCMP process, the diamond workpieces were polished with diamond abrasive so as to present the same roughness. Then, the surface roughness was traced to evaluate the material removal in PCMP. Additionally, it also prevents uneven polishing due to the repeating paste in the PCMP. As shown in the figure, the diamond surface roughness lowers gradually during the duration of the 8 h polishing operation. The slurries with photocatalyst have stronger oxidizability than those without photocatalysts. Therefore, observation shows that the photocatalysis properties of P25 or 5 nm titanium dioxide are better. Many hydroxyl radical ·OHs are created in the slurry, which is critical in the removal of diamond. Among the three groups, the polishing slurry prepared with P25 titanium dioxide exhibits the best polishing results, and the diamond's surface roughness decreases from Ra 33.6 nm to Ra 2.6 nm. Furthermore, P25 titanium dioxide's photocatalytic activity is relatively stable, so it can maintain its photocatalyzed capability for a long time.

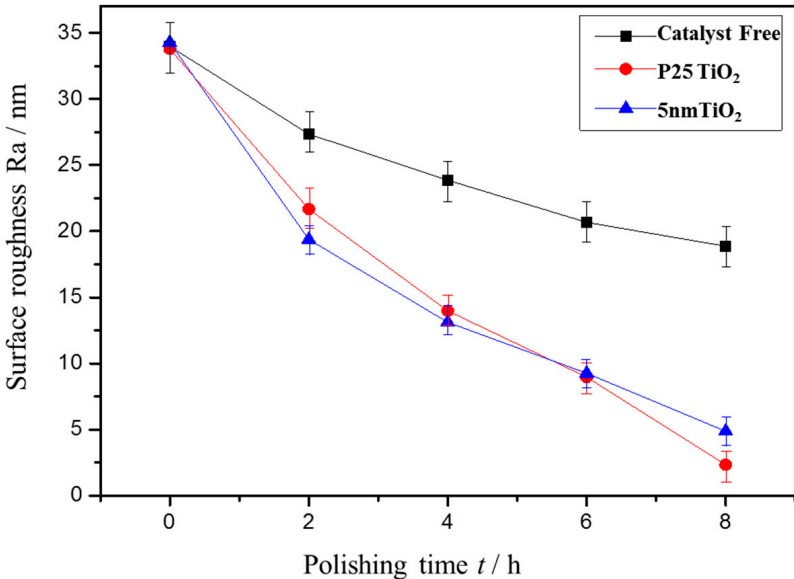

**Figure 16.** The surface roughness of diamond varying with polishing time.

The diamond surface morphology before and after polishing is shown in Figure 17. As can be seen from the figure, the diamond's initial surface contains multiple residual mechanical scratches. However, after being polished without photocatalyst for eight hours, the surface becomes smoother, with minor scratches. The majority of mechanical scratches are removed when the diamond workpiece that has undergone mechanical lapping is polished for eight hours with a slurry of 5 nm $TiO_2$ or P25 $TiO_2$. Therefore, a smooth surface can be observed on a diamond workpiece. The surface morphology of a diamond workpiece measured with AFM in a 5 μm × 5 μm shown in Figure 18 presents a small surface roughness of about Ra1.6 nm after polishing with 5 nm $TiO_2$.

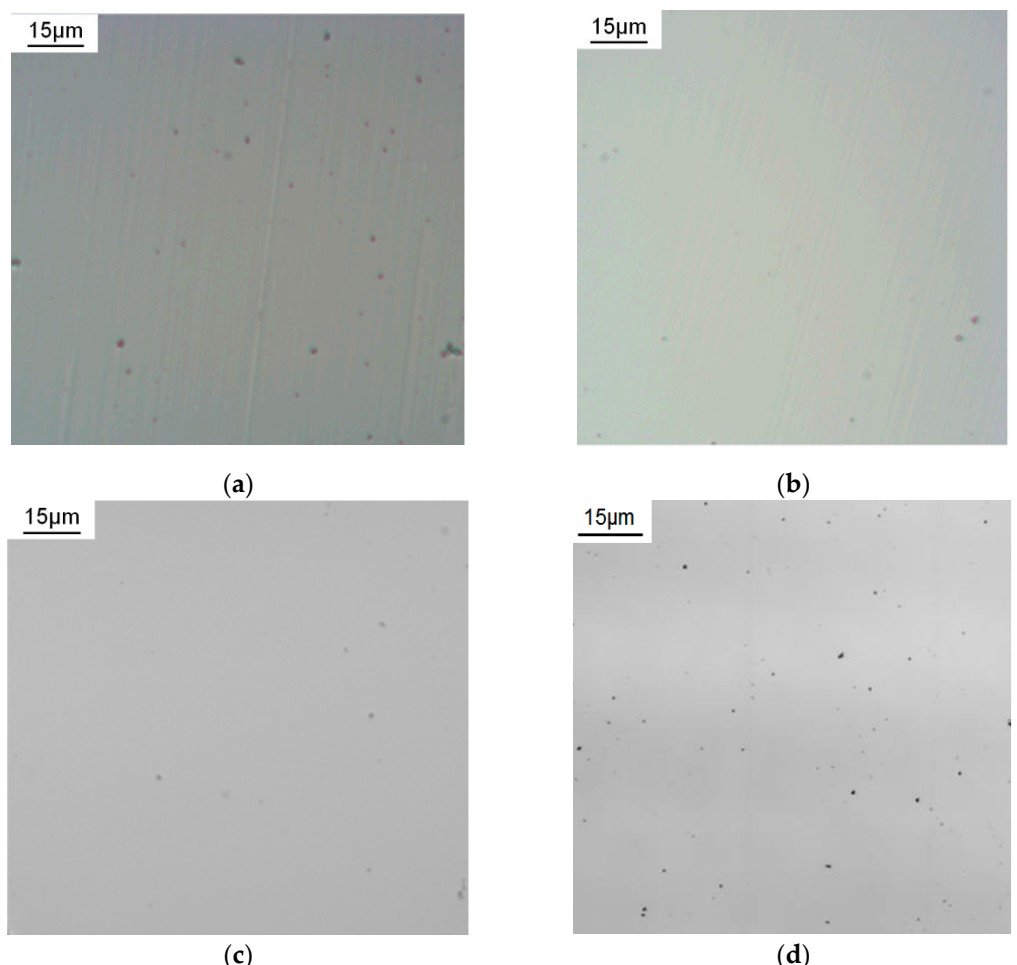

**Figure 17.** The diamond surface morphology before and after polishing. (**a**) Initial surface; (**b**) polishing without photocatalyst for 8 h; (**c**) polishing with 5 nm $TiO_2$ for 8 h; (**d**) polishing with P25 $TiO_2$ for 8 h.

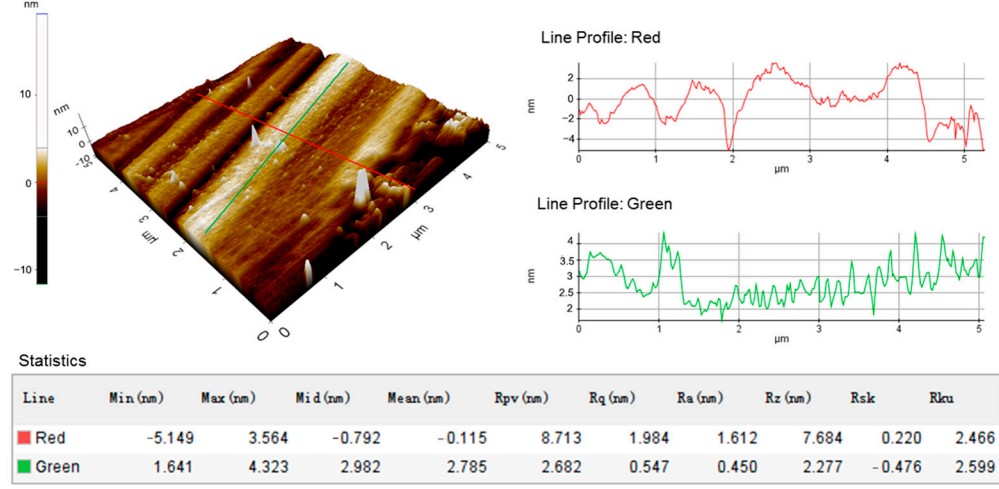

**Figure 18.** The three-dimensional morphology of diamond after polishing with 5 nm $TiO_2$.

## 4. Conclusions

In this paper, a novel method for polishing diamonds is provided that makes use of the hydroxyl radical ·OH produced on the photocatalyst particles' surface when they are exposed to UV light. The production of a polishing slurry for the proposed PCMP is also covered. The following are some insightful conclusions:

(1) The removal of diamonds via photocatalysis-assisted chemical mechanical polishing is effective as it utilizes hydroxyl radical ·OH as an oxidant in the slurry. The ECMP slurry contains a photocatalyst, abrasive, electron capture agent, pH regulator and dispersant to achieve an optimal effect.

(2) The maximum ORP is present in the 5 nm $TiO_2$ and P25 $TiO_2$ solutions. By incorporating $H_2O_2$ and $H_3PO_4$ into the slurry and exposing it to UV light, the oxidizability of the slurry increases. Both $H_2O_2$ and $H_3PO_4$ are neither detrimental to the environment nor to humans; however, $K_2FeO_4$ decomposes more easily than $H_2O_2$.

(3) The ORP of slurry and the oxidation of the diamond can both be improved by acid condition. $TiO_2$ powder and $H_2O_2$ can be used to boost slurry conductivity, but, as the $TiO_2$ and $H_2O_2$ concentrations reach a particular threshold, the gain in conductivity stops.

(4) Methyl orange is an appropriate reagent for determining whether a slurry is oxidizable because the UV light will cause the yellow color to disappear after 60 min.

(5) Both P25 $TiO_2$ and 5 nm $TiO_2$ exhibit strong photocatalysis properties. Surface roughness can be decreased from Ra 33.6 nm to Ra 2.6 nm in 8 h using a slurry containing P25 $TiO_2$. Moreover, PCMP can be used to remove mechanical scratches from diamond surfaces.

**Author Contributions:** Conceptualization, Z.Y. and J.S.; methodology, Z.Y. and Y.Z.; experiments, Z.Y. and H.D.; formal analysis, J.Z.; investigation, Q.W.; writing—original draft preparation, Z.Y. and H.D.; writing—review and editing, Z.Y. and J.S.; visualization, Q.W.; supervision, Z.Y. and Y.Z.; project administration, Z.Y.; funding acquisition, Z.Y. and J.S. All authors have read and agreed to the published version of the manuscript.

**Funding:** This research was funded by the National Natural Science Foundation of China, China (grant number 52275455), Liaoning Revitalization Talents Program, China (grant number XLYC2007133) and Open Foundation of State Key Laboratory of Superabrasives, China (grant number 2022-0-43-020).

**Data Availability Statement:** All relevant data are within the paper.

**Acknowledgments:** This paper is supported by the National Natural Science Foundation of China, China (grant number 52275455), Liaoning Revitalization Talents Program, China (grant number XLYC2007133) and Open Foundation of State Key Laboratory of Superabrasives, China (grant number 2022-0-43-020). We thank the associate editor and the reviewers for their useful feedback, which improved this paper.

**Conflicts of Interest:** The authors declare no conflict of interest.

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
