# Peer review of "A New Slurry for Photocatalysis-Assisted Chemical Mechanical Polishing of Monocrystal Diamond"

_machines, doi:10.3390/machines11060664_

Round 1

Reviewer 1 Report

This manuscript proposes photocatalysis chemical mechanical polishing on diamond. Follows are the comments on this work.

1.       The English of this manuscript is not well organized, which should be corrected and honed by a native English speaker. For instance, the first sentence of 1. Introduction is false, indicating the carelessness of the authors. In addition, there are typos existing in this manuscript, such as “chemical chemical polishing or mechanical polishing”, etc.

2.       There should be a blank between a digit and its followed unit, such as in Abstract, 5 nm and 2.6 nm, etc.

3.       smaller than 1 μm” is correct, and micrometer is not um.

4.       In Abstract, what is ORP? When an abbreviation firstly appears, it should be written with full words, and otherwise it will confuse the readers.

5.       Chemical mechanical polishing is presented in this work, while recent significant progresses on them are lost. In this regard, follows should be added in Introduction. Chemical mechanical polishing (CMP) is an effective method to achieve atomic surface globally (Applied Surface Science 253 (2007) 9137-9141). However, traditional CMP usually employs toxic and corrosive ingredients, resulting in the pollution to the environment (COLLOIDS AND SURFACES A-PHYSICOCHEMICAL AND ENGINEERING ASPECTS 586 (2020) 124293). To overcome this challenge, novel green CMP is developed for copper (Applied Surface Science, 467-468 (2019) 5-11), alloys (Materials Letters 122 (2014) 252-255) and diamond (Applied Surface Science, 564 (2021) 150431). Using the green CMP, high-performance surfaces are fabricated for the use in semiconductor, aerospace and microelectronics industries (Colloids and Surfaces A: Physicochemical and Engineering Aspects, 662 (2023) 131000). The most important is that these studies are a great contribution to the conventional CMP and manufacturing, effectively eliminating the pollution to the environment (Nanoscale, 2020, 12 (2020) 22518-22526).

6.       In Fig. 1, the first letter of phrases should be a capital, rather than a usual letter.

7.       Figure 2 is blurry, which should be replaced by clear SEM ones. Additionally, the scale bar and unit should be redrawn.

8.       Equation 1 should cite references.

9.       In Fig. 15, a space is lost between a bracket and its subsequent words.

10.    In Fig. 16, the vertical coordinate should be Surface roughness Ra, and standard deviation is absent for the experimental results. For a test, at least three experiments should be repeated, and then an average value and standard deviation could be obtained. In fact, all the measurements including Figs. 9, 10, 11, 12, 13 and 14, should contain standard deviation. Once result is not convinced.

11.    A schematic diagram of CMP polishing mechanism should be provided and added in text.

The English of this manuscript is not well organized, which should be corrected and honed by a native English speaker. For instance, the first sentence of 1. Introduction is false, indicating the carelessness of the authors. In addition, there are typos existing in this manuscript, such as “chemical chemical polishing or mechanical polishing”, etc.

Author Response

Dear Editor,

Thanks for your prompt reply and the kindly comments of the reviewers. We have carefully considered the recommendations and suggestions of the reviewers for our manuscript and below address those comments. Changes to the manuscript are all indicated below.

The following is a point-to-point response to the reviewers’comments.

Respond to Reviewer #1:

  1. The English of this manuscript is not well organized, which should be corrected and honed by a native English speaker. For instance, the first sentence of 1. Introduction is false, indicating the carelessness of the authors. In addition, there are typos existing in this manuscript, such as “chemical chemical polishing or mechanical polishing”, etc.

 Thanks very much for your careful comments. We revised the manuscript carefully again and corrected the language errors of this manuscript.   

  1.  There should be a blank between a digit and its followed unit, such as in Abstract, 5 nm and 2.6 nm, etc.

Thanks very much for your careful comments. We revised the manuscript carefully again and added a blank between the digit and unit.   

  1.  “smaller than 1 μm” is correct, and micrometer is not um.

Thanks very much for your important comment. We revised the manuscript carefully.   

  1.  In Abstract, what is ORP? When an abbreviation firstly appears, it should be written with full words, and otherwise it will confuse the readers.

Thanks very much for your important comment. The ORP is the abbreviation of “oxidation-reduction potential”. We revised it.   

  1.  Chemical mechanical polishing is presented in this work, while recent significant progresses on them are lost. In this regard, follows should be added in Introduction. Chemical mechanical polishing (CMP) is an effective method to achieve atomic surface globally (Applied Surface Science 253 (2007) 9137-9141). However, traditional CMP usually employs toxic and corrosive ingredients, resulting in the pollution to the environment (COLLOIDS AND SURFACES A-PHYSICOCHEMICAL AND ENGINEERING ASPECTS 586 (2020) 124293). To overcome this challenge, novel green CMP is developed for copper (Applied Surface Science, 467-468 (2019) 5-11), alloys (Materials Letters 122 (2014) 252-255) and diamond (Applied Surface Science, 564 (2021) 150431). Using the green CMP, high-performance surfaces are fabricated for the use in semiconductor, aerospace and microelectronics industries (Colloids and Surfaces A: Physicochemical and Engineering Aspects, 662 (2023) 131000). The most important is that these studies are a great contribution to the conventional CMP and manufacturing, effectively eliminating the pollution to the environment (Nanoscale, 2020, 12 (2020) 22518-22526).

Thanks very much for your important comment. We revised the manuscript and added some references into the introduction of CMP as following,   

Although it is faster than ion beam sputtering, the diamond surface may become contaminated because of plasma heating. The cost is relatively significant, and the plasma size restricts the sample size. Chemical mechanical polishing (CMP) is an effective method to achieve atomic surface globally and so the most promising technique for large-area diamond is chemical mechanical polishing [20, 21]. To extract the diamond atom by chemical reaction or tribochemical reaction, it employs either the molten mixed salt containing NaNO3, KNO3 and KOH or produced slurry with strong oxidants containing CrO3、KMnO4、H2O2、K2Cr2O7 [22]. However, the traditional CMP slurrys usually contains toxic and corrosive ingredients, resulting in the pollution to the environment [23].

To overcome this chanllenge of traditional slurry and increase the material removal rate, strong oxidants like K2FeO4 and Fenton reagent are used to prepare polishing slurry [24, 25]. These studies are a great contribution to the conventional CMP and manufacturing, effectively eliminating the pollution to the environment [26, 27]. However, the polishing effect might be negatively impacted by storage failure of this type of polishing slurry with ahigh oxidant. Waste liquid treatment and recovery are highly expensive, and discharging waste liquid after polishing can seriously harm the environment and human. Furthermore, the polishing parameters can be used to actively after the mechanical action in polishing process, but the chemical action is much more difficult to control. Given this, the current study proposes a photocatalysis-assisted chemical mechanical polishing (PCMP) approach to accomplish the untra-smooth polishing of large-area diamonds via the production of a polishing slurry for the proposed PCMP.

  1. Shi Z.; Jin Z.; Guo X.; et al. Insights into the atomistic behavior in diamond chemical mechanical polishing with OH environment using ReaxFF molecular dynamics simulation. Computational Materials Science 2019, 166, 136-142.
  2. Wang Y.; Zhao Y. and Wang J. Modeling the effects of cohesive energy for single particle on the material removal in chemical mechanical polishing at atomic scale, Applied Surface Science 2007, 253(23), 9137-9141.

22.Charrier G.; Lévy S.; Vigneron J.; et al. Electroless oxidation of boron-doped diamond surfaces:comparison between four oxidizing agents;Ce4+,MnO4,H2O2 and S2O82−. Diamond and related materials 2011, 20(7), 944-950.

  1. Zhou J.; Niu X.; Wang Z.; et al. Roles and mechanism analysis of chitosan as a green additive in low-tech node copper film chemical mechanical polishing, Colloids and Surfaces A: Physicochemical and Engineering Aspects 2020, 586: 124293.

24.Yuan Z.; Jin Z.; Zhang Y.; et al. Chemical mechanical polishing slurries for chemically vapor-deposited diamond films. Journal of manufacturing science and engineering 2013, 135(4).

  1. Liao L.; Zhang Z.; Meng F.; , et al. A novel slurry for chemical mechanical polishing of single crystal diamond, Applied Surface Science 2021, 564:150431.
  2. Li Y.; Zhang Z.; Shi, C.; et al. Density functional theory analysis and novel green chemical mechanical polishing for potassium dihydrogen phosphate, Colloids and Surfaces A: Physicochemical and Engineering Aspects 2023, 662: 131000.
  3. Xie W.; Zhang Z.; Liao L.; et al. Green chemical mechanical polishing of sapphire wafers using a novel slurry, Nanoscale 2020, 12, 22518-22526.

  1.  In Fig. 1, the first letter of phrases should be a capital, rather than a usual letter.

Thanks very much for your kind remind. We change the first letter to capital letter.

  1.  Figure 2 is blurry, which should be replaced by clear SEM ones. Additionally, the scale bar and unit should be redrawn.

Thanks very much for your important comment. We change Figure 2 in more clear figures and redraw the unit.

  1.  Equation 1 should cite references.

Thanks very much for your important comment. We change Figure 2 in more clear figures and redraw the unit.

  1.  In Fig. 15, a space is lost between a bracket and its subsequent words.

Thanks very much for your important comment. We added the space in Figure 15 and Figure 17.

  1.  In Fig. 16, the vertical coordinate should be Surface roughness Ra, and standard deviation is absent for the experimental results. For a test, at least three experiments should be repeated, and then an average value and standard deviation could be obtained. In fact, all the measurements including Figs. 9, 10, 11, 12, 13 and 14, should contain standard deviation. Once result is not convinced.

Thanks very much for your important comment. We added standard deviation to the Figure 16. Other figures are most process data, which are affected by recording time. It is difficult to add the standard deviation. We choose the data which can reflect the experimental process.  

  1.  A schematic diagram of CMP polishing mechanism should be provided and added in text.

Thanks very much for your important comment. We added the schematic diagram of CMP polishing mechanism.

Reviewer 2 Report

In the paper entitled "A New Slurry for Photocatalysis Assisted Chemical Mechanical Polishing Monocrystal Diamond”, the authors used the PCMP method to polish the diamond, achieving a smooth surface. Here are some questions which may help improve the manuscript as follows:

(1)     Several grammar mistakes and nonstandard writings need to be carefully revised. For example, "chemical vapor deposition (CVD) method had break through" should be "chemical vapor deposition (CVD) method had broken through”; "ml" should be “mL”; “PH value will” should be “pH value will”; There should be a space between the number and the unit; There should be a space between “Fig.” and the following number. In the 3.2 section, "The initial addition of 0.2ml phosphoric acid significantly increased the ORP value, reaching 501mv 436mv". Is it 501 mV or 436 mV? Why does the author use “P25 TiO2” instead of “25 nm TiO2”? Please read through the manuscript and correct the errors.

(2)     The font of the legend in some figures seems too small. Please improve the figures. In addition, Figure 10 is the same as Figure 9, and Figure 10 is inconsistent with the description in the manuscript. Please check it.

(3)     Please add error bars to the experimental data.

(4)     In the Principle, methods and experiments section, the authors stated, “the abrasives of alumina, silicon carbide and diamond abrasives are used.” However, in the following experimental results, only the diamond abrasive was tested. Please explain. Moreover, please provide the exact slurry composition used in the study.

(5)     In the Principle, methods and experiments section, Figure 3 exhibits the oxidation characterization test for detecting oxidation-reduction potential (ORP), pH value and conductivity of slurry. Please provide more details about the measurement principle of ORP and the device including the producer and the model. Moreover, why did the authors use ORP instead of the redox potential? Why did the authors measure the conductivity? What is the relationship between the material removal rate and the surface roughness and the conductivity?

(6)     As shown in Figure 8, “the addition of TiO2 can not increase the ORP when there is H2O2 and H3PO4 in slurry”. Please explain.

(7)     As stated, "When diamond powder is added into slurry on the condition of UV photocatalysis, the carbon of diamond can be oxidized into CO or CO2". Please give the chemical reactions. Meanwhile, please provide experimental evidence or literature support for the chemical reactions.

(8)     As shown in Figure 15, “After UV irradiation for 30min, methyl orange solution 306 showed obvious discoloration”. As stated, “the fade time can reflect the oxidizability of PCMP slurry”. Did the authors experiment without UV irradiation and compare the fade time? Because H2O2 itself is a strong oxidizer without UV irradiation.

(9)     As shown in Figure 17, there are defects on the diamond surface after polishing, especially with P25 TiO2. It is hard to describe it as ultra-smooth. Please explain.

(10) In Figure 16, how did the authors measure the surface roughness? Please add the corresponding 3D surface morphologies after Figure 16. Meanwhile, the PCMP is mainly used to improve the polishing efficiency. Please provide the material removal rates of the slurries with and without catalyst in Figure 16.

Several grammar mistakes and nonstandard writings need to be carefully revised. 

Author Response

Dear Editor,

Thanks for your prompt reply and the kindly comments of the reviewers. We have carefully considered the recommendations and suggestions of the reviewers for our manuscript and below address those comments. Changes to the manuscript are all indicated below.

The following is a point-to-point response to the reviewers’comments.

Respond to Reviewer #2:

  1. Several grammar mistakes and nonstandard writings need to be carefully revised. For example, "chemical vapor deposition (CVD) method had break through" should be "chemical vapor deposition (CVD) method had broken through”; "ml" should be “mL”; “PH value will” should be “pH value will”; There should be a space between the number and the unit; There should be a space between “Fig.” and the following number. In the 3.2 section, "The initial addition of 0.2ml phosphoric acid significantly increased the ORP value, reaching 501mv 436mv". Is it 501 mV or 436 mV? Why does the author use “P25 TiO2” instead of “25 nm TiO2”? Please read through the manuscript and correct the errors.

Thanks very much for your careful comments. We revised the manuscript carefully again and corrected the language errors of this manuscript. We changed “ml” to “mL”, “PH” to “pH”. We add a space between the number and the unit.

  1.  The font of the legend in some figures seems too small. Please improve the figures. In addition, Figure 10 is the same as Figure 9, and Figure 10 is inconsistent with the description in the manuscript. Please check it.

Thanks very much for your careful comments. We revised the manuscript carefully again and improved the figures. We corrected the Figure 10.  

  1.  Please add error bars to the experimental data.

Thanks very much for your important comment. We added error bars to the Figure 17 (Figure 16). Other figures are most process data, which are affected by recording time. It is difficult to add the standard deviation. We choose the data which can reflect the experimental process.  

  1.   In the Principle, methods and experiments section, the authors stated, “the abrasives of alumina, silicon carbide and diamond abrasives are used.” However, in the following experimental results, only the diamond abrasive was tested. Please explain. Moreover, please provide the exact slurry composition used in the study.

Thanks very much for your important comment. Diamond abrasives are used foe the rough lapping of diamond workpiece, which can reduces the surface roughness rapidly. And then the diamond workpiece is polished with PCMP method. During the PCMP operation, the abrasives of alumina, silicon carbide and boron carbide are used. However, alumina abrasive is more usefull for the PCMP. So alumina is the main abrasive for PCMP.  The slurry for PCMP contains 0.5g of P25 TiO2 particles, 6g of aluminum oxide abrasives and 3ml of H2O2 in every 100 mL water. The polishing rotational speed and pressure are 60 r/min and 1.09 MPa, respectively.  

  1.  In the Principle, methods and experiments section, Figure 3 exhibits the oxidation characterization test for detecting oxidation-reduction potential (ORP), pH value and conductivity of slurry. Please provide more details about the measurement principle of ORP and the device including the producer and the model. Moreover, why did the authors use ORP instead of the redox potential? Why did the authors measure the conductivity? What is the relationship between the material removal rate and the surface roughness and the conductivity?

Thanks very much for your important comment. The ORP was measured with AZ86505 oxidation-reduction potentiometer (AZ Instrument Corporation). The ORP values were recorded when different photocatalyst, electron capture agent and pH regulator were added into solutions. Oxidation-reduction potential is same meaning to redox potential. The oxidation-reduction potential is common used in references, so we use ORP in this study. During the PCMP operations, we find the conductivity varying with various amounts of photocatalyst and abrasives. And it will affect the measurement of oxidation-reduction potential. Additionally, electric is also used in PCMP experiments to increase the rate of diamond removal in our following study. So the conductivity was recorded in the experiments. There is no direct relationship between conductivity and material removal rate and surface roughness. But the conductivity of solution can affect the ORP value of slurry. And the macrooxidability can be reflected by ORP value. the conductivity of solution will affect the microoxidability of a TiO2 particles. Unfortunately, we still have no proper method to measure the oxidability of a particle area. So we hope to find that the conductivity can establish a relationship between the macrooxidability of sollution and the microoxidability of a TiO2 particles.       

  1.  As shown in Figure 8, “the addition of TiO2 can not increase the ORP when there is H2O2 and H3PO4 in slurry”. Please explain.

Thanks very much for your important comment. the addition of H2O2 and H3PO4 can significantly increase the oxidizability of the slurry. But if there is already H2O2 and H3PO4 in the slurry, the addition of TiO2 can not increase the ORP. The main reason is that the gas film produced on the surface of electrode will affect the measurment of ORP when H2O2 and H3PO4 were added into the solution. So the microoxidability of a TiO2 particles is difficult to measure during the experiment. The measured ORP values will smaller than actual ORP values of a particle area. 

  1.  As stated, "When diamond powder is added into slurry on the condition of UV photocatalysis, the carbon of diamond can be oxidized into CO or CO2". Please give the chemical reactions. Meanwhile, please provide experimental evidence or literature support for the chemical reactions.

Thanks very much for your important comment. Diamond is unstable in chemical thermodynamics, it can be oxidized into CO or CO2 by strong oxidant. This study used the hydroxyl radicals ·OH with a strong oxidability to remove diamond according the principle. We added the chemical reactions in the text. The mechanisms how diamond is oxidized into CO or CO2 will be studied in our next manuscript. We do a lot of simulation and experiment to analyze the oxidation of diamond. According to the experiments, the pH of solution declines due to the dissolution of CO or CO2.

  1.  As shown in Figure 15, “After UV irradiation for 30min, methyl orange solution 306 showed obvious discoloration”. As stated, “the fade time can reflect theoxidizability of PCMP slurry”. Did the authors experiment without UV irradiation and compare the fade time? Because H2O2 itself is a strong oxidizer without UV irradiation.

Thanks very much for your important comment. Actually we did the various experiments with or without UV irradiation. The oxidizability of PCMP slurry with UV exposure is higher than that of without UV exposure although there is H2O2 in solution.

  1.   As shown in Figure 17, there are defects on the diamond surface after polishing, especially with P25 TiO2. It is hard to describe it as ultra-smooth. Please explain.

Thanks very much for your important comment. The diamond surface has some micro holes in some area due to the growth of CVD method. However, the diamond surface polishing with P25 has no mechanical scratches in comparison with mechanical polishing or polishing without photocatalyst. The surface roughness of diamond surface is affected by mechanical scratches significantly.

  1.  In Figure 16, how did the authors measure the surface roughness? Please add the corresponding 3D surface morphologies after Figure 16. Meanwhile, the PCMP is mainly used to improve the polishing efficiency. Please provide the material removal rates of the slurries with and without catalyst in Figure 16.

. Thanks very much for your important comment. The surface roughness was measured Mitutoyo SJ-412 surface roughness measuring instrument. The instrument can not produce 3D surface morphologies. Because the material removal rates are very small (usually dozens of nanometer), it is difficult to measure the material removal rate. So we usually observe the change of surface morphologies or surface roughness to judge the material removal rate. For example, we can roughly judge the material removal according to 3D surface morphology of rough diamond in different polishing stage as following figure. But the roughness can not reflect the polishing effects because there too much initial rough asperities on diamond surface.  

   11.  Several grammar mistakes and nonstandard writings need to be carefully revised.

Thanks very much for your careful comments. We revised the manuscript carefully again and corrected the language errors of this manuscript. We also polished the language of the manuscript carefully.

Reviewer 3 Report

This manuscript entitled “A New Slurry for Photocatalysis Assisted Chemical Mechanical Polishing Monocrystal Diamond” reports a novel approach employing a photocatalyst for chemical mechanical polishing (CMP) of mechanically hard and chemically stable diamond material. While using oxidation promotion through photocatalysis is interesting, there are issues with the clarity of the experimental results and the consistency of the graphs. As a result, it becomes challenging to discern whether the observed effects of CMP are primarily attributed to photocatalytic properties or mechanical action. Supplemental data is needed on the variation of ORP values with photocatalysts, evidence that this enhances the properties of CMP, and how the photocatalyst altered the surface properties of the diamond. It is also unclear whether the addition of TiO2 photocatalyst enhances the diamond CMP properties by a chemical reaction or is simply an effect of increasing the abrasive concentration.

Considering these concerns, it is recommended to reject the manuscript in its current state. It needs that further revisions are necessary to address the issues above adequately. Additionally, when re-submitting a paper later, authors are encouraged to use a professional English editing service to proofread their manuscript.

Author Response

Dear Editor,

Thanks for your prompt reply and the kindly comments of the reviewers. We have carefully considered the recommendations and suggestions of the reviewers for our manuscript and below address those comments. Changes to the manuscript are all indicated below.

The following is a point-to-point response to the reviewers’comments.

Respond to Reviewer #3:

  1. This manuscript entitled “A New Slurry for Photocatalysis Assisted Chemical Mechanical Polishing Monocrystal Diamond” reports a novel approach employing a photocatalyst for chemical mechanical polishing (CMP) of mechanically hard and chemically stable diamond material. While using oxidation promotion through photocatalysis is interesting, there are issues with the clarity of the experimental results and the consistency of the graphs. As a result, it becomes challenging to discern whether the observed effects of CMP are primarily attributed to photocatalytic properties or mechanical action. Supplemental data is needed on the variation of ORP values with photocatalysts, evidence that this enhances the properties of CMP, and how the photocatalyst altered the surface properties of the diamond. It is also unclear whether the addition of TiO2 photocatalyst enhances the diamond CMP properties by a chemical reaction or is simply an effect of increasing the abrasive concentration.Considering these concerns, it is recommended to reject the manuscript in its current state. It needs that further revisions are necessary to address the issues above adequately. Additionally, when re-submitting a paper later, authors are encouraged to use a professional English editing service to proofread their manuscript.

Thanks very much for your careful comments. We revised the manuscript carefully one by one sentence and make it easier to understand. We had studied the CMP of diamond for many years. We ever used a lot of oxidants including KMnO4、(NH4)2S2O8、K2FeO4、CrO3、K2Cr2O7 and 30ml H2O2 to diamond. In order to increase the rate of material removal, the concentration of these oxidant is very high. However it will rise many problems. The evaporation of slurry with strong oxidant will etch equipment and be harmful to human body. Our nose bleed for many times due to the CMP experiment. So this basis, we proposed the PCMP method. Therefore the concentration of the oxidant can decrease. In our previous experiment, it is resultful to remove diamond and silicon carbide with PCMP method. According to the Fig.17, the slurry with photocatalyst can reduce the diamond surface roughness faster than the slurry without photocatalyst. Additionally, the material removal rate of diamond is very low in comparison with other material. The obtain of experiment data is much difficult than other material. The process may undergoes the following chemical reaction. The mechanisms how diamond is oxidized into CO or CO2 will be studied in our next manuscript. We sincerely hope you can consider our manuscript. Thank you very much.

Round 2

Reviewer 1 Report

All the comments have been revised and corrected, and now it is appropriate for publication.

Author Response

Thanks very much for your important comments. We revised the manuscript carefully again.

Reviewer 2 Report

 There are still some questions not being properly addressed.

(1)     Several grammar mistakes and nonstandard writings need to be carefully revised. For example, "ml" should be “mL”; There should be a space between the number and the unit; “an untra-smooth surface” should be “an ultra-smooth surface”; “Therfore” should be “Therefore”; Why does the author use “P25 TiO2” instead of “25 nm TiO2”? Please read through the manuscript and correct the errors.

(2)     Please compare the polishing performance of the abrasives of alumina and diamond, and briefly explain the advantages of alumina.

(3)     As the authors stated, there is no direct relationship between the material removal rate and the surface roughness and the conductivity. It would be better not to use the conductivity data. Moreover, As shown in Figure 9, “the addition of TiO2 can not increase the ORP when there is H2O2 and H3PO4 in slurry”. As explained by the authors, the measured ORP cannot accurately reflect the oxidation capacity. If it is true, why did the authors measure the ORP? Please explain.

(4)     As stated, "Through UV photocatalysis, the diamond powder can oxidize its carbon into CO or CO2 when combined with slurry." Please provide experimental evidence or literature support for the chemical reactions.

(5)     Please provide the experimental results of methyl orange without UV irradiation and compare the fade time.

(6)     As shown in Figure 18, there are defects on the diamond surface after polishing, especially with P25 TiO2. It is hard to describe it as ultra-smooth (surface roughness<1 nm). As the authors explained, the diamond surface has some micro holes in some area due to the growth of CVD method. Please provide evidence.

(7)     If there are no 3D surface morphologies, please provide the original 2D surface morphologies after Figure 17. Meanwhile, the PCMP is mainly used to improve the polishing efficiency. If the material removal rates are low. The authors can either polish more time to measure the material removal or roughly evaluate the material removal rate by the change of the surface roughness.

Several grammar mistakes and nonstandard writings need to be carefully revised. 

Author Response

Dear Editor, Thanks for your prompt reply and the kindly comments of the reviewers. We have carefully considered the recommendations and suggestions of the reviewers for our manuscript and below address those comments. Changes to the manuscript are all indicated below. The following is a point-to-point response to the reviewers’comments. Respond to Reviewer #2: 1. Several grammar mistakes and nonstandard writings need to be carefully revised. For example, "ml" should be “mL”; There should be a space between the number and the unit; “an untra-smooth surface” should be “an ultra-smooth surface”; “Therfore” should be “Therefore”; Why does the author use “P25 TiO2” instead of “25 nm TiO2”? Please read through the manuscript and correct the errors. Thanks very much for your careful comments. We revised the manuscript carefully again. We checked the manuscript and changed all “ml” to “mL”. We add a space between all number and the unit. We read through and corrected the errors. 2. Please compare the polishing performance of the abrasives of alumina and diamond, and briefly explain the advantages of alumina. Thanks very much for your careful comments. Diamond and alumina are common abrasives in lapping and chemical mechanical polishing. Diamond abrasives are the most hard abrasives, which are usually used in the process of lapping or mechanical polishing. They are not suitable for CMP and PCMP because diamond grits in slurry can be oxidized along with diamond workpiece. Additionally, diamond abrasives are easy to cause scratches on workpiece. The material removal rate will significantly decline with the decrease of diamond abrasive size. We ever did a lot of experiments to analyze the effectiveness of abrasives. It is difficult to remove diamond material by using only alumina abrasive. In this study, in order to verify that the effect of chemical action in PCMP on the material removal of diamond. Therefore, the diamond abrasives are used as the abrasive of mechanical lapping for diamond rough treatment. Alumina abrasives are used in PCMP slurry in this study because they can not be oxidized by the oxidant in slurry. 3. As the authors stated, there is no direct relationship between the material removal rate and the surface roughness and the conductivity. It would be better not to use the conductivity data. Moreover, As shown in Figure 9, “the addition of TiO2 can not increase the ORP when there is H2O2 and H3PO4 in slurry”. As explained by the authors, the measured ORP cannot accurately reflect the oxidation capacity. If it is true, why did the authors measure the ORP? Please explain. Thanks very much for your important comment. In traditional CMP slurry, the oxidant usually presents ionic condition dispersing in slurry. So it is easy to volatilize to be harmful to the environment. In ECMP, the oxidants are produced on the surface of nano particles. It can not affect the conductivity of slurry. The conductivity of slurry are mainly affected by the amount of nano particles and other additive. Although there is no direct relationship between the material removal and the conductivity, the conductivity of slurry will provide a condition foe the measurement of ORP. Additionally, the conductivity of slurry will provide a convenience for the electric assisted ECMP. We revised the description again carefully as following. Fig. 9 illustrates how ORP varies with TiO2 concentration in the presence of UV radiation. ORP increases as TiO2 concentration increases because that more hydroxyl radicals are produced on the surface of TiO2 particles, which also increase the conductivity of the slurry. The conductive solution is benificial for the measurement of ORP. However, over much TiO2 particles in slurry will decline the absorbtion of UV light due to that the slurry becomes non-transparent when there are a lot of TiO2 particles. Ihat is why the increase of ORP is not obvious when the concetration of TiO2 particles is high, especially when there is already H2O2 and H3PO4 in the slurry. Additionally, it should be noted that the addition of H2O2 and H3PO4 can significantly increase the oxidizability of the slurry. The introduction of acidic condition is useful to increase the oxidizability of slurry. It should be noted that the gas film produced on the surface of electrode will affect the measurement of ORP when H2O2 and H3PO4 were added into the solution. The measured ORP values will smaller than actual ORP values of a particle area. Nano TiO2 particles is conductive. So the slurry conductivity will change when nano TiO2 particles are added into the slurry. Fig. 14 illustrates how the conductivity of the solution varies as different quantities of TiO2 are added into 50 mL water. The conductivity of a solution seems to be improved by the addition of a catalyst. When the slurry with nano TiO2 particles are placed under the radiation of UV light. The conductivity of slurry can not change obviously no matter how much nano TiO2 particles are added. It indicates that the photocatalysed reaction usually ocurrs on the surface of nano TiO2 particles, which do not affect the conductivity of slurry. The conductivity of slurry mainly affected by the amount of nano TiO2 particles. Additionally, the pH of slurry continuously increases after adding nano TiO2 particles. It implies that the slurry undegoes some chemical reaction. It should be noted that excess of TiO2 powder will reduce the absorbance of UV light, and that will decrease the efficiency of the photocatalytic reaction. 4. As stated, "Through UV photocatalysis, the diamond powder can oxidize its carbon into CO or CO2 when combined with slurry." Please provide experimental evidence or literature support for the chemical reactions. Thanks very much for your careful comments. We added literature support for the chemical reactions. We also did a lot of experiments and molecular dynamics simulation to analysis the oxidation of diamond carbon during PCMP, and will present it in next study in detail. 37.Liu W.T.; Xiong Q.; Lu J.B.; Wang X.H.; et al. Tribological behavior of single crystal diamond based on UV photocatalytic reaction. Tribology International 2022, 175, 107806. 38.Zhang L.; Hamers R. J.; Photocatalytic reduction of CO2 to CO by diamond nanoparticles. Diamond and Related Materials 2017, 78, 24-30. 5. Please provide the experimental results of methyl orange without UV irradiation and compare the fade time. Thanks very much for your careful comments. The oxidation experiments of methyl orange was analyzed in our previous study [1]. So in this study, we emphasize the fade process of methyl orange. [1] Zewei Yuan, Yan He, Xingwei Sun, and Quan Wen. UV-TiO2 photocatalysis-assisted chemical mechanical polishing 4H-SiC wafer, Materials and Manufacturing Processes, 2018, 33(11): 1214–1222. 6. As shown in Figure 18, there are defects on the diamond surface after polishing, especially with P25 TiO2. It is hard to describe it as ultra-smooth (surface roughness

Reviewer 3 Report

This manuscript entitled “A New Slurry for Photocatalysis-assisted Chemical Mechanical Polishing Monocrystal Diamond” reports the effect of TiO­2 catalyst for PCMP. Compared to the previous manuscript, it describes the role and mechanism of TiO2 catalyst in PCMP slurry and is well revised logically. The manuscript would be suitable for publication in ‘Machine’ after minor revisions. Detailed comments are as follows;

1.       According to the authors, the graph in Fig. 8 should show a higher ORP when the slurry containing the catalyst is treated with UV light, but instead it shows a lower ORP. It is necessary to verify that the graph has not been altered.

2.       Despite the authors' careful review, there are still some typos. For example, "ahigh" in line 88, ".." in the last sentence of line 248, ",The" in the first sentence of line 252, etc. Authors are encouraged to correct their manuscripts for typographical errors.

3.       The study used diamond particles as the abrasive for the PCMP. However, it is questionable whether the micrographic image of the alumina powder in Figure 3 is necessary. It is recommended that the authors consider removing the data for the alumina powder and keeping only the data for the diamond particles, the abrasive used in the experiment.

Author Response

Dear Editor, Thanks for your prompt reply and the kindly comments of the reviewers. We have carefully considered the recommendations and suggestions of the reviewers for our manuscript and below address those comments. Changes to the manuscript are all indicated below. The following is a point-to-point response to the reviewers’comments. Respond to Reviewer #3: This manuscript entitled “A New Slurry for Photocatalysis-assisted Chemical Mechanical Polishing Monocrystal Diamond” reports the effect of TiO2 catalyst for PCMP. Compared to the previous manuscript, it describes the role and mechanism of TiO2 catalyst in PCMP slurry and is well revised logically. The manuscript would be suitable for publication in ‘Machine’ after minor revisions. Detailed comments are as follows. Thanks very much for your evaluation on our manuscript. 1.According to the authors, the graph in Fig. 8 should show a higher ORP when the slurry containing the catalyst is treated with UV light, but instead it shows a lower ORP. It is necessary to verify that the graph has not been altered. Thanks very much for your careful comments. We changed a figure so as to indicate the ORP values of different photocatalysts varying with the ultraviolet irradiation time. We revised the description as following. The oxidizability of a solution can be assessed using the ORP value and in water, it varies with different photocatalysts as illustrated in Fig. 8. The ORP values of 5 nm TiO2 are highest among various photocatalysts and have a obvious increase when the slurry is placed under the UV light. The ORP values of 10 nm TiO2, P25 TiO2 and ZnO are smaller than 5 nm TiO2 but larger than 20 nm TiO2 distinctly. It indicated the photocatalytic performance is not only affected by the size of photocatalyst particles but also is related to the crystal form of photocatalysts. P25 is a type of mixed crystal TiO2. It possesses a higher ORP values than 20 nm TiO2 although the typical size of P25 TiO2 particles is about 25 nm. The lowest ORP is seen in pure water because of its low oxidizability. Additionally, the ORP values have a remarkable increase when H2O2 is added into the solution. It is owing to that the H2O2 increases the conductivity of the solution and prevent the recombination of electrons and holes due to its high oxidizability. 2.Despite the authors' careful review, there are still some typos. For example, "ahigh" in line 88, ".." in the last sentence of line 248, ",The" in the first sentence of line 252, etc. Authors are encouraged to correct their manuscripts for typographical errors. Thanks very much for your careful comments. We revised the manuscript carefully again. 3.The study used diamond particles as the abrasive for the PCMP. However, it is questionable whether the micrographic image of the alumina powder in Figure 3 is necessary. It is recommended that the authors consider removing the data for the alumina powder and keeping only the data for the diamond particles, the abrasive used in the experiment. Thanks very much for your careful comments. We deleted the Figure 3.
